# Experimental Breast Phantom Imaging with Metamaterial-Inspired Nine-Antenna Sensor Array

**DOI:** 10.3390/s18124427

**Published:** 2018-12-14

**Authors:** Mohammad Tariqul Islam, Md Samsuzzaman, Md Tarikul Islam, Salehin Kibria

**Affiliations:** Centre of Advanced Electronic and Communication Engineering, Faculty of Engineering and Built Environment, Universiti Kebangsaan Malaysia, 43600 Bangi, Selangor, Malaysia; kibriasalehin@gmail.com

**Keywords:** microwave imaging, breast tumor, metamaterial loaded antenna, ultrawideband antenna, homogenous phantom

## Abstract

An experimental system for early screening of a breast tumor is presented in this article. The proposed microwave imaging (MI) system consists of a moveable array of nine improved negative-index metamaterial (MTM)-loaded ultrawideband (UWB) antenna sensor with incorporation of a corresponding SRR (split-ring resonator) and CLS (capacitively loaded strip) structure, in a circular array, the stepper motor-based array-mounting stand, the adjustable phantom hanging platform, an RF switching system to control the receivers, and a personal computer-based signal processing and image reconstruction unit using MATLAB. The improved antenna comprises of four-unit cells along one axis, where an individual unit cell integrates a balancing SRR and CLS pair, which makes the antenna radiation omnidirectional over the operating frequencies. The electrical dimensions of this proposed antenna are 0.28*λ* × 0.20*λ* × 0.016*λ*, measured at the lowest operating frequency of 2.97 GHz as the operating bandwidth of this is in between 2.97–15 GHz (134.82% bandwidth), with stable directional radiation pattern. SP8T 8 port switch is used to enable the eight receiver antennas to sequentially send a 3–8.0 GHz microwave signal to capture the backscattered signal by MATLAB software. A low-cost realistic homogeneous breast phantom with tumor material is developed and measured to test the capability of the imaging system to detect the breast tumors. A post-processing delay-multiply-and-sum (DMAS) algorithm is used to process the recorded backscatter signal to get an image of the breast phantom, and to accurately identify the existence and located area of multiple breast tumor tissues.

## 1. Introduction

An appealing and inspiring interest in the field of electromagnetic waves and antennas in medical applications, with microwave systems, has arisen in recent years. Microwave imaging (MI) is a promising candidate for breast tumor detection [1,2,3,4,5] as differences between electrical properties are identified using a microwave sensor. In the microwave imaging system, the power is radiated over an antenna sensor, and another one or pair of sensors receive the scattered power. The scattered signals are further processed to detect the unwanted malignant tissues. Ultrawideband has the advantage of deep penetration and higher resolution features. The conventional MI systems that are reported in the literature are proposed for the detection of a tumor inside human breast tissue [6,7,8].

There is still a significant challenge to implement a metamaterial structure-inspired antenna in MI systems. Various categories of antennas are proposed and used for breast phantom measurements, for instance, Pyramidal horn antenna [9,10], Vivaldi antenna [11,12,13,14,15], CPW antenna [16], metamaterials, and EBG antenna [17], array antenna [18], the slotted antenna [19,20,21], and Fourtear antenna [22]. The metamaterial embraces a non-natural electromagnetic construction which has negative permeability/permittivity covering a perceptible frequency range. As a result of having prodigious possibilities, the metamaterial created a new approach in MI applications by producing microwave sensors like an antenna. In 1968, Veselago predicted an engineered material, theoretically, that showed negative permeability and permittivity at the same time [23]. Pendry proposed a metamaterial having split ring resonator (SRR) structure [24] in 1999 and, eventually, Smith performed the demonstration and justification of metamaterial perception in 2000 [25]. In the meantime, various left-handed metamaterials are reported using different structures, like fishnet structures [26], SRRs [27], multiple SRRs [28], layouts of transmission line [29], double-sided SRRs [30], spiral SRRs [31], H-shaped pairs periodic arrays [32], cut wire pairs [33], SRR pairs [34], double-bowknot-shaped resonators [35], and broadside-coupled SRRs [36]. The spectrum and range of the unit cell is curbed due to having a narrow frequency band and, therefore, being not easy to implement in the antenna, as well as fabricate. As time goes on, the areas for metamaterial application are expanding as these complications are overcome. A planar-pattern metamaterial was reported in [37,38], where authors formed a coupled capacitive-inductive circuit by modifying the radiating elements. The overall dimensions of the metamaterial (MTM) [37] planar patterned antennas were 28 mm × 32 mm, covering the bandwidth from 5.3 to 8.5 GHz (46.37% fractional bandwidth) with around 4 dBi of average peak gain using high-cost Rogers substrate. With the average gain of 5.42 dBi, the antenna [38] covers the operating bandwidth from 3.85 to 15.62 GHz (120.90% fractional bandwidth), and the dimensions were 27.6 mm × 31.8 mm using high-cost F4BM-2 substrate. The planar patterned MTM antenna proposed in this paper has achieved more than 134.82% (2.97 to 15 GHz) fractional bandwidth, and greater than 3 dBi average peak gain across the operating band through the dimensions of 27.5 mm × 19.40 mm in the low-cost FR4 substrate, which has better performance than [37,38]. The authors of [37,38] have designed an antenna only, but not used it for any application. Besides, both articles claimed metamaterial of their unit cell, but they did not present any permeability or permittivity characteristics of their unit cell for claiming MTM antenna. A metamaterial-inspired antenna for ultrawideband (UWB) applications was reported in [39], but the dimensions are larger than the proposed antenna and do not cover the UWB band (3.1–10.6 GHz). An anisotropic zero-index metamaterial loaded antenna was proposed for microwave imaging applications where the authors did not clarify the imaging process [40]. Although the antenna covers a wide range of frequencies, the dimensions are too large to implement in portable imaging systems.

In this paper, the authors have presented the design and development of an imaging system that can be utilized for breast imaging. The developed imaging system can be utilized to detect the tumor and its position in the case of breast imaging. In this system, a metamaterial-inspired UWB antenna array of nine prototypes is implemented to send and receive signals. Metamaterial unit cells were used to enhance the impedance matching and radiation performance of each antenna prototype. The nine antennas can rotate in their track to work as a multistatic system. A lab-made breast phantom has also been fabricated and measured through dielectric coaxial probe kit. The MATLAB-based software and imaging system hardware allows finishing the data collection procedure within three minutes for 8 × 50 scanned channels. After processing the collected microwave backscattered data, positive image results had been acquired to detect the high dielectric multiple tumor objects embedded in the phantoms by using the DMAS algorithm [41].

## 2. Metamaterial Unit Cell Design Layout

The design of the UWB antenna starts with the designing of a metamaterial unit cell. The primary target is to determine a unit cell having resonance properties in between the frequency range of 3.1 to 10.6 GHz. Numerous methods are used to design metamaterial structure, including split ring resonators (SRRs) [24,25]. The authors choose the most popular SSR structure for developing the unit cell. The structure of SRR contains two loops, where smaller loops belongs to a bigger loop, and opposite ends are slotted [24]. The magnetically resonant SRR structure produces a vertical magnetic field that is responsible for creating negative permeability. The controls of resonant behavior of the unit cell are attained using splits to the ring that familiarize capacitance. The metamaterial, which shows both negative permeability and permittivity concurrently, are named double-negative (DNG) metamaterials. They can also be single-negative materials, like epsilon-negative (ENG) or mu-negative (MNG). The metamaterial that has double- or single-negative properties can attain better performance in numerous essential applications, such as polarization rotators, invisibility cloaking, SAR reduction, and many more. Figure 1a represents the proposed unit cell of rectangular SSR. Figure 1b shows the simulation geometry of the unit cell in Computer Simulation Technology (CST) simulation software [42]. The structure is printed on a low-cost FR4 substrate which dielectric constant of 4.6 and height of 1.6 mm, to achieve the resonance of the unit cell within 3.1–10.6 GHz, and two capacitive loaded strips (CLSs) are accumulated for the modification of the SSR unit cell. Here, I-shaped strip line impersonator as extended metallic line, and CLS, act as electric dipoles [43]. The concurrent electric and magnetic resonance among the SRR is possible, due to the implementation of a combined structure that allows creating a vertical magnetic field, and CLS resonates over an equivalent electric field [44]. The design specification of the unit cell is summarized in Table 1.

The simulation of the metamaterial unit cell is carried out using Computer Simulation Technology (CST) software based on the finite-difference time domain (FDTD). The S-parameters are observed from the simulation results. Figure 1b represents the simulation setup. The unit cell is located inside two waveguide ports on both sides of the x-axis, and the electromagnetic wave is excited along this axis. Along the walls vertical to the y-axis, a perfect electrical boundary condition is applied, and the magnetic conducting boundary is applied through the z-axis. For the simulation, frequency domain solver is applied by applying normalized matched impedance of 50 Ω. The S parameters (both reflection and transmission coefficient) are presented in Figure 2. Multiple transmission peaks are observed at the left-handed band that occurs at 5.2 and 10 GHz. The proposed SRR structure magnetic resonance is better than the reported overlap and self-oriented SRRs [27]. The fundamental useful parameters are extracted from *S*_21_ and *S*_11_ using the Nicolson–Ross–Weir approach [28,45] that includes permeability *μ_r_*, permittivity *ε_r_*, and refractive index *n_r_*.

These following equations are accomplished separately, consistent with
(1)$$\varepsilon_{r}=\frac{2}{jk_{0}d}\times \frac{1-V_{1}}{1+V_{1}},$$
(2)$$\mu_{r}=\frac{2}{jk_{0}d}\times \frac{1-V_{2}}{1+V_{2}},$$
(3)$$n_{r}=\sqrt{\varepsilon_{r}\mu_{r}},$$
(4)$$V_{1}=S_{21}+S_{11},$$
(5)$$V_{2}=S_{21}-S_{11},$$
where:


$k_{0}=\omega /c$


ω=2πf, angular frequency

d= slab thickness

c= speed of light

By following these Equations (1)–(5), all the effective parameters are calculated. The permeability, permittivity, and the refractive index are shown in Figure 3. The frequency covering the negative region is enlisted in Table 2. It is identified that the resonance frequencies from 5.30–7.90 GHz and 9.50–10.25 GHz, in both permeability and permittivity, are negative values, and then the structure can be listed in DNG metamaterial. The refractive indices are also initiated to be negative for the frequencies of 7.36–10.48 GHz and 12.85–13.15 GHz.

## 3. Antenna Sensor Design with MTM

The schematic layout of the optimized MTM UWB antennas’ sensor design layout is displayed in Figure 4. The proposed design consists of a patch and ground plane, which is printed on 1.6 mm thick low-cost epoxy resin fiber FR4 substrate with dielectric constant 4.6. The UWB antenna patch is constructed with a feed line and a triangular strip patch. The four MTM identical unit cells are connected with the triangular copper strip along the y-axis. On the other hand, the partial ground plane is printed on the other side of the substrate with a rectangular and circular slot. To enhance the antenna performance, including impedance bandwidth, gain, and radiation, the MTM unit cell is used. A 50 Ω SMA connector is attached to the edge of the antenna to connect patch and ground plane. After optimization of the proposed antenna, the final design parameter is presented in Table 3.

After inserting the MTM unit cell, the VSWR (Voltage Standing Wave Ratio) performance of the realized antenna is displayed in Figure 5. From this figure, it can be observed that without inserting the unit cell with the triangular strip and partial ground plane, the antenna achieves −10 dB impedance from 4.20 to 15 GHz, which does not cover the UWB frequency range (3.1–10.60 GHz) range. To shift the −10 dB resonance from 4.20 to 3.10 GHz, we have attached one, two, three, and four MTM unit cells with the triangular strip. The four-unit cell MTM antenna achieves the desired UWB frequency range regarding VSWR. The peak gain effect of the inserting MTM unit cell is also depicted in Figure 6. After attaching four identical MTM unit cells, the antenna achieved comparatively better gain across the desired UWB operating band.

The magnitude and vector surface current distribution of the realized antenna is presented in Figure 7 at 3.25, 6.50, and 9.5 GHz. It can be observed that the current density flow dominates around the feed and triangular strip in the low-frequency band. On the other hand, the current flows density is dominant around the MTM unit cell with the fed and triangular strip, which plays an essential role in generating resonance and achieving UWB frequency band. At 6.50 GHz, the first and fourth MTM unit cell is greatly affected with the current flow, which ensures the UWB performance. On the other hand, at 9.50 GHz, the second and third unit cell with the triangular strip was excited more to achieve a wide bandwidth.

## 4. Antenna Sensor Performance Validation

The numerical investigation has been performed in 3D electromagnetic simulator HFSS (High-Frequency Structure Simulator) and CST. After manual optimization of the realized antenna sensor dimension parameter through the simulation software, the design was fabricated in the UKM Lab-based PCB prototype machine LPKF. The fabricated prototype of the designed antenna is shown in Figure 8, top and bottom view. The VSWR was measured in UKM microwave lab using the N5227A (10–67 GHz) network analyzer. The far-field characteristics of the realized antenna were measured by Satimo near the field measurement facilities of the UKM microwave lab, which is shown in Figure 8c. The numerical and measured VSWR of the realized antenna is depicted in Figure 9. The numerical VSWR in HFSS less than 2 is about 3.10–15 GHz and 2.8–15 GHz in the CST simulation software. On the other hand, a measured VSWR was achieved from 2.97 to 15 GHz, which fully covered the UWB band. The simulated results significantly matched with the measured results. Figure 10 illustrated the simulated and measured 2D polar radiation pattern of the realized antenna (XZ plane and YZ plane) at 3.25, 6.50, and 9.5 GHz, respectively. The radiation pattern also displayed the co-polarization and cross-polarization results. It can be noted that the antenna exhibits a shape pattern as in Figure 8 in the XZ plane, and an omnidirectional pattern in the YZ plane, which is a dipole-like radiation pattern. The cross-polarization is comparatively low with respect to polarization. Figure 11 depicts the simulated and measured peak realized gain of the antenna. From this figure, it can be stated that the antenna achieves more than 3 dB peak gain across the desired operating band.

## 5. Microwave Imaging Setup with Multiple Receiver

The breast imaging system is designed to evaluate the imaging performance of detecting breast tumor using a realistic lab-based breast phantom. The architecture and different component of the proposed experimental breast imaging system are depicted in Figure 12a. The proposed microwave imaging system consists of an antenna array, (9 antennas, one for transmitting and eight for receiving the ultrawideband signals), the stepper motor-based antenna mounting stand, the flexible phantom hanging platform, an RF switching system to control the receivers, and the personal computer-based signal processing and image reconstruction unit. The used antennas are mounted on an adjustable transparent plastic rotational table. The nine plastic sticks are installed on a rotating platform with an SD02B controlled stepper motor. The breast phantom is placed inside the antenna array using a hanging platform and scanned using the nine MTM-based UWB antenna array. The gap between the antenna and the phantom is maintained at 140 mm. The mechanical rotation platform can rotate the antenna array in polar coordinates from 0 to 2π around the breast phantom using the stepper motor. The antennas are connected to a GaAs MMIC SP8T (9 Port) non-reflective positive control switching network using low-loss coaxial cables. The received signal from all eight receivers is collected by switching the receiving antennas. The data (S21, S31, S41, S51, S61, S71, and S81) are collected at each 7.2°, and 50 equal points, covering the total 360°. The imaging system uses an Agilent E8358A vector network analyzer (VNA) microwave transceiver. The port 1 of VNA generates microwave signals through the transmitting antenna in the frequency domain, and transmits it to the breast phantom. The backscattered signal is received by the remaining eight antennas by another port via the SP8T RF switch, and sent to the image processing unit. All these devices and electromechanical circuits, related to the data acquisition process, are controlled by a PC-controlled Arduino Uno control circuit which is connected to the personal computer through the USB port. The VNA is also connected to the PC via a GPIB port, and data are received for further processing. The collected data are processed by a laptop using the DMAS imaging algorithm, which reconstructs the image of the breast interior to detect and localize a tumor object.

## 6. Phantom Fabrication with Imaging Setup

The lab-based breast phantom used for the imaging system are discussed in the phantom development and measurement procedure in [15]. The radii of the phantoms are 55 mm (phantoms A, B, and C) and 60 mm (phantom D and phantom E) which are displayed in Figure 12b. The tumor diameter and height are 10 mm and 40 mm, respectively. Holes were drilled manually in the appropriate positions of the phantoms, and the tumor material was poured in to form the tumors. This resulted in some tumors being placed at a slight offset. Four phantoms were constructed with a different tumor configuration. Phantom A is a homogeneous phantom without any tumors. It serves as the control test, that should appear blank due to the rotation subtraction method. Phantom B is constructed by adding a tumor 25 mm away from the center of the structure. This will be the primary test where the imaging system has to detect only a single target. Usually, DAS-based methods excel in such situations, as reflections off only one target have to be considered. A more challenging phantom C is created by replicating phantom B and adding another tumor on the opposite side of the center at an approximately equal distance. The presence of multiple targets can cause numerous internal reflections to misguide the imaging system. Thus, DMAS is utilized in the proposed system as it rewards higher coherence by multiplying individual pairs of delayed signals and adding them together as a form of coherence measurement. This quantity is used to scale the results obtained from the conventional DAS. Finally, Phantom D is constructed slightly larger at a radius of 60 mm as it is intended for a four-tumor objects. The tumor objects are also placed slightly further apart, at 35 mm at 90-degree intervals around the center, to maintain the structural integrity of the phantom. Mapping four different targets is usually not attempted or presented in most examples in the literature, as it is often difficult for DAS-based techniques, which were developed mostly as single target detection methods. A more realistic heterogenous phantom with four layers (skin, fat, gland, and tumor) is presented as phantom E in Figure 12b, which has more human-like dielectric properties and construction. Figure 13 represents the measured and targeted dielectric constant and electrical conductivity of each material of the phantoms against frequency. The dielectric constant and conductivity of homogenous phantom in Figure 13a,b is identical with the targeted curves. In addition, the properties of each material (skin, fat, gland, and tumor) of heterogenous phantom shown in Figure 13c,d demonstrates the accurate measurement properties with the targeted values. Therefore, the results of the phantom properties in this paper have the more realistic characteristics of the real human breast to be tested with the microwave imaging system efficiency. After developing and measuring the breast phantom, it is placed inside the antenna array, and the attenuation and reflection of breast tissues are considered on the performance of antennas, which is shown in Figure 14. The VNA parameters are set as IF (intermediate frequency) bandwidth 100 Hz, 10 dBm output power, and 3.0 to 8 GHz frequency range with 201 discrete points. The transmitting antenna transmits the microwave pulse towards the breast phantom, and the receiving eight antennas receive the scattering signals reflected from the phantom after every 7.2 degrees. To reduce air interference, the entire system is calibrated over the operating frequency using SOLT (Short-Open-Load-Thru) calibration (3.5 mm Agilent 85052 D) kit. The losses and permittivity of water are much higher than air, and real breast tissues, which create higher attenuation and reflection in the air–skin interface, are considered as the experimental method.

## 7. Imaging Results Analysis and Discussions

After capturing the backscattered signal through the developed imaging system, the data were processed to identify the breast phantom tumor object. By using the developed experimental setup, the complex frequency domain S-parameter $\Gamma \left(\varphi_{n},f_{m}\right)$ data are captured, where *m* = 1, 2, …, *M*, and *n* = 1, 2 …, *N* represent the angular positions of each rotation. The reflected parameter mostly presents the shallow depths under the skin layer, as signals are bounced off the opposite side of the breast phantom have to travel through the phantom twice, and are significantly attenuated. Thus, antennas with very low inherent return loss are ideal for detecting weak reflected signals. Methods to remove the reflections from the skin are critical for detecting scattered signals from inside the phantom, since reflection from the air–skin interface is orders of magnitude stronger than reflections from the tumors. Several techniques, like matching liquids [46] and rotation subtraction [47], have been employed. Matching liquids have dielectric properties, like skin, thus allowing maximum power coupling to the phantom’s internal structure. Such designs require antennas that can operate when immersed in the matching liquid. However, matching liquid causes uncomfortable breast compression during measurements, and increases the overall weight and complexity of the system. Rotation subtraction relies on a comparison between an original illumination and at least one rotated illumination [48]. In such systems, the antenna array is placed around the region of interest. Once the data are recorded for the original illumination, the array is rotated around the phantom to get offset data.

In this study, the Γ(*φ_n_*, *f_m_*) is separated into two matrices by *n* being odd and even, or Γ_odd_(*φ_l_*, *f_m_*) and Γ_even_(*φ_l_*, *f_m_*), respectively, where *l* = 1, 2, …, *N*/2. Thus, Γ_odd_(*φ_l_*, *f_m_*) can be considered original illumination, and Γ_even_(*φ_l_*, *f_m_*) is the ‘offset’ illumination of the first matrix. Finally, rotation subtraction is implemented by just calculating the difference between the two matrices.
Γ(*φ_l_*, *f_m_*) = Γ_odd_(*φ_l_*, *f_m_*) − Γ_even_(*φ_l_*, *f_m_*)(6)

Fourier transform is used to convert the reflection coefficient from frequency domain to time domain for each antenna.
(7)$$S\left(\varphi_{l},t_{k}\right)=exp\left\{\left[D_{k\times m}\right]\right\}\times \Gamma \left(\varphi_{l},f_{m}\right)=\left[\begin{array}{ccc} S\left(\phi_{1},t_{1}\right) & \ldots & S\left(\phi_{N/2,}t_{1}\right)\\ \vdots & \ddots & \vdots \\ S\left(\phi_{1},t_{m}\right) & \ldots & S\left(\phi_{N/2},t_{k}\right) \end{array}\right],$$
where
(8)$$f_{m}=f_{1}+\left(m-1\right)\left(f_{h}-f_{1}\right)/\left(M-1\right),$$
(9)$$\Gamma \left(\varphi_{l},f_{m}\right)=\left[\begin{array}{ccc} \Gamma \left(\phi_{1},f_{1}\right) & \ldots & \Gamma \left(\phi_{N/2,}f_{1}\right)\\ \vdots & \ddots & \vdots \\ \Gamma \left(\phi_{1},f_{M}\right) & \ldots & \Gamma \left(\phi_{N/2},f_{M}\right) \end{array}\right],$$
and
(10)$$\left[D_{k\times m}\right]=\left[\begin{array}{ccc} j\omega_{1}t_{1} & \ldots & j\omega_{m}t_{1}\\ \vdots & \ddots & \vdots \\ j\omega_{1}t_{m} & \ldots & j\omega_{m}t_{k} \end{array}\right].$$

Here, *ω_m_* denotes the angular velocity and *k* indicates the equal distant points. 

Subsequently, the data in the S matrix was processed using the delay-multiply-and-sum (DMAS) algorithm for the precise reconstruction of the image [49]. The pictures of the developed breast phantoms with tumor are shown in Figure 15. We are using cylindrically symmetric homogeneous and heterogeneous phantom (excluding the tumors), and the phantom is placed at the center of the rotation axis. Thus, the skin reflections are nearly identical for all observations, and any discrepancies between the even and odd sets must be due to scattered signals from the internal structure of the phantom. A continuous green circle is drawn on the final imaging results to indicate the phantom surface, which is shown in Figure 15. Figure 15a is mostly blank as expected, due to the homogeneity of phantom A. Small insignificant specks of noise appear at the surface of phantom A, possibly due to cracks on its exterior. Figure 15b shows a single point of high contrast to the fat material as white, along with some lower intensity clutter below it. The high contrast location is recognized as the center of the tumor, and the clutter can be attributed to imperfect insertion of tumor, resulting in minor cracks in the fat being filled with tumor material. Figure 15c clearly shows two separate clutters, indicating the presence of two tumors. However, the upper tumor in Figure 15c is slightly closer to the center than the lower tumor. Upon re-examination of phantom C, it was noted that one tumor was placed closer. The discrepancy can be observed in Figure 15c. Finally, Figure 15d shows one high contrast clutter, most likely caused by the tumor nearest to the skin. The other three tumors also appear, but only as low contrast clusters. Since they are buried more in-depth, the reflected signals face more attenuation, resulting in less contrast. However, the presence of four distinct clutters indicates the detection of all four tumors. Figure 15e depicts the imaging results of the heterogeneous phantom with two tumor objects, which is shown in 12b. From this imaging picture, it can be noted that two tumors like object have detected with some other object, due to the different layers. Table 4 represents the comparison of the different imaging system with proposed system regarding the antenna, the number of an element, the number of scanning position, static or fixed antennas, measurement of time or frequency domain, and method of imaging algorithm. Initially, our system emphasizes the compact lab-based antenna design and nine-antenna array-based imaging system developed for testing the lab-based breast phantom (homogeneous and heterogeneous), with and without tumor condition, through the MATLAB-based DMAS imaging algorithm. Good imaging results have achieved from the system, which are highly desirable for performing nondestructive biomedical detections.

## 8. Conclusions

A microwave system for breast tumor imaging is presented in this article. The designed imaging system comprises an array of nine metamaterial loaded antennas sensor that can work across the UWB band. Simulation and measured results of this proposed antenna demonstrate the operating bandwidth of 2.97 to 15 GHz with reflection coefficient <−10 dB, and stable peak gain across the operating band. The proposed antenna displays a stable omnidirectional radiation pattern which is the primary requirement for microwave imaging. A suitable SP8T device is used to enable the eight receiver antennas to 50 rotated position, to send reflected microwave signals, whereas the reflected backscattered signals were recorded PNA (Performance Network Analyzer) using the MATLAB-based software architecture. Several lab-based breast phantoms that match the dielectric properties of real breast tissues with breast tumor were fabricated and measured to test the validity of the imaging system. After collecting the data, a post-processing DMAS algorithm was used to significantly detect the breast tumor existence and location inside the developed phantom. Successful images have been presented, which recognize a single tumor object or multiple tumor objects, identified using the measured data by the developed imaging system.

## Figures and Tables

**Figure 1 sensors-18-04427-f001:**
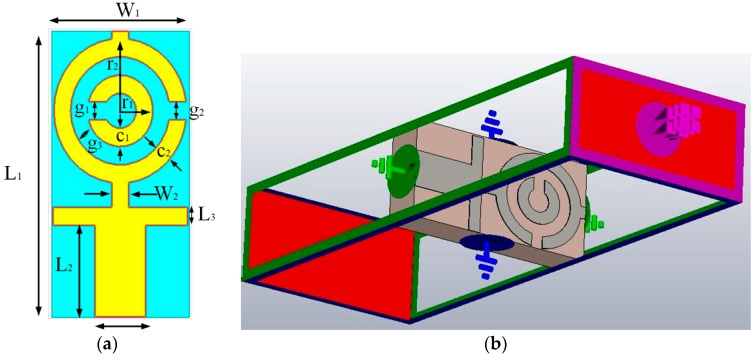
(**a**) The unit cell front side; and (**b**) simulation geometry [42].

**Figure 2 sensors-18-04427-f002:**
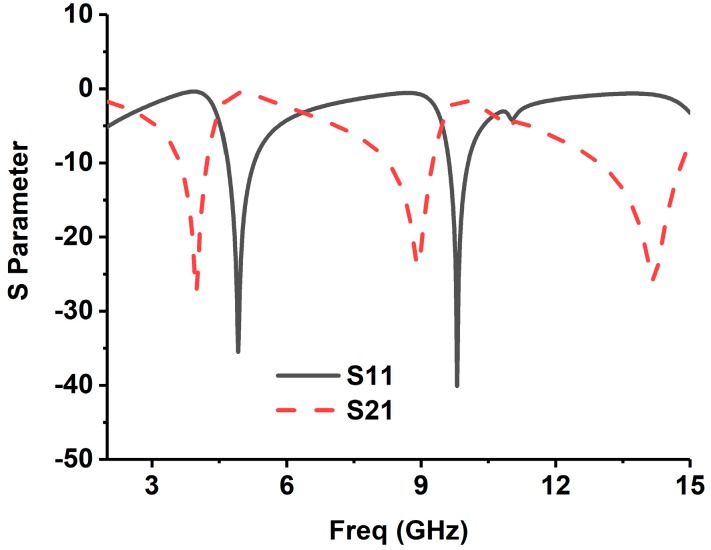
The magnitude of S-parameters (*S*_11_ and *S*_21_) for the reported unit cell (Figure 1).

**Figure 3 sensors-18-04427-f003:**
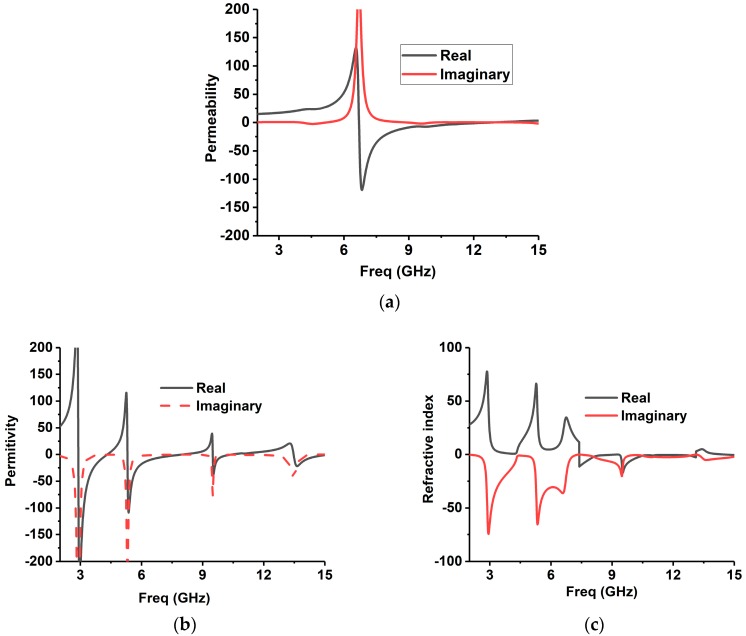
The proposed unit cell (**a**) permeability, (**b**) permittivity, and (**c**) refractive index.

**Figure 4 sensors-18-04427-f004:**
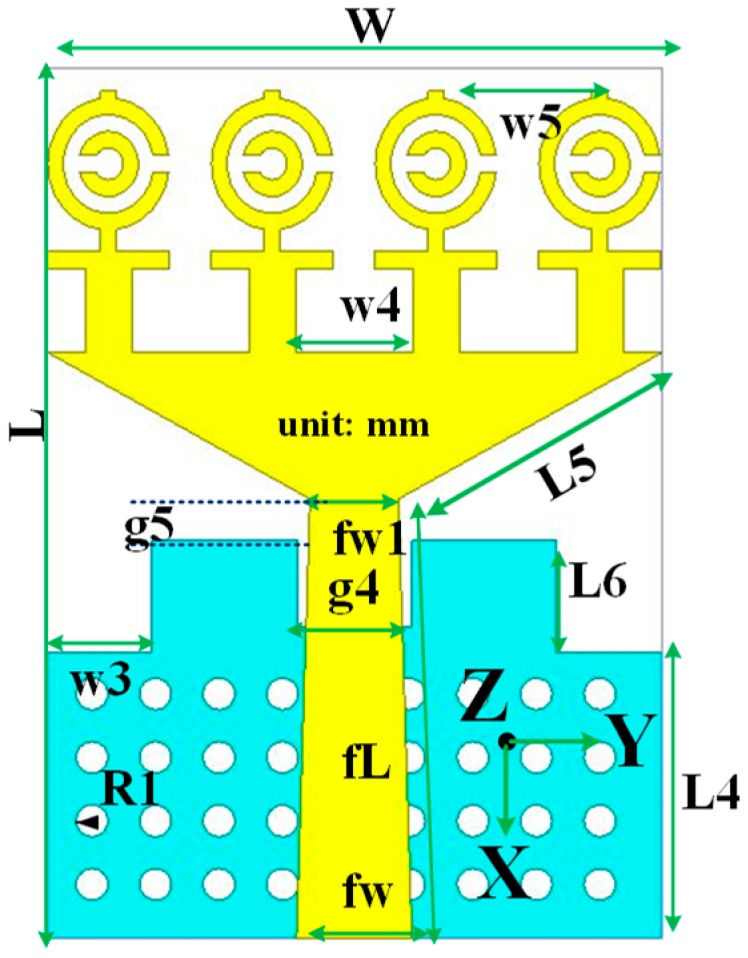
The reported metamaterial (MTM) antenna layout.

**Figure 5 sensors-18-04427-f005:**
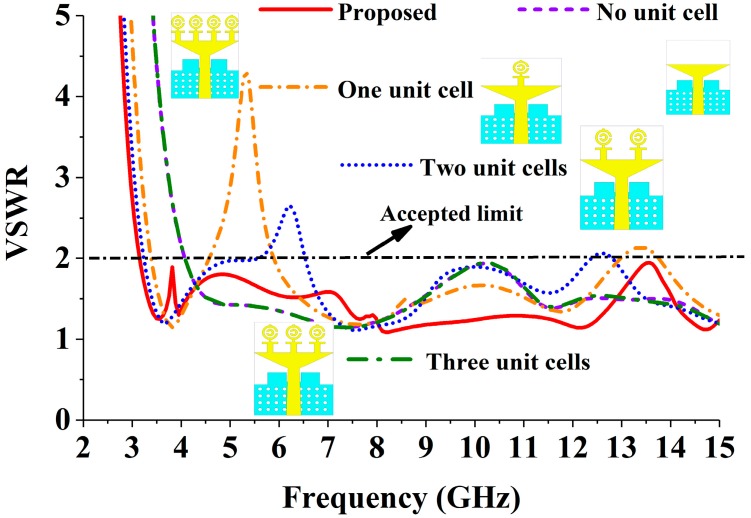
The effects of the unit cell of the radiating patch on the VSWR.

**Figure 6 sensors-18-04427-f006:**
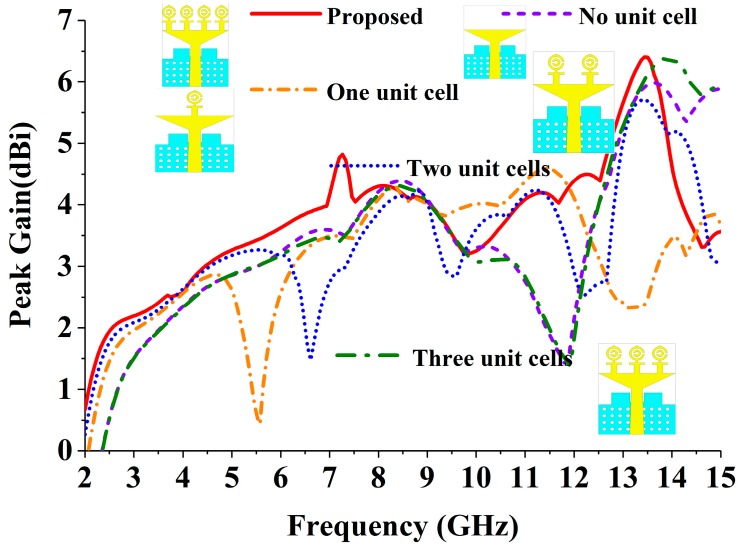
The effects of the unit cell of the radiating patch on the peak gain.

**Figure 7 sensors-18-04427-f007:**
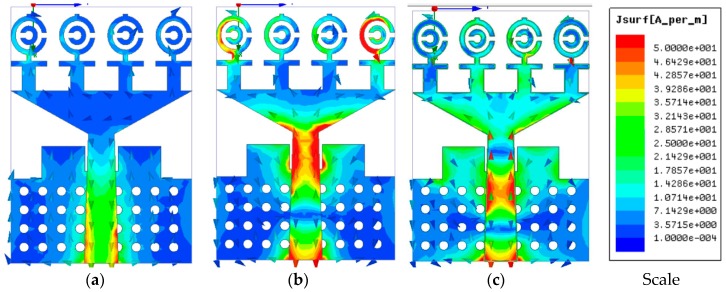
The surface current distribution at (**a**) 3.25, (**b**) 6.5, and (**c**) 9.5 GHz.

**Figure 8 sensors-18-04427-f008:**
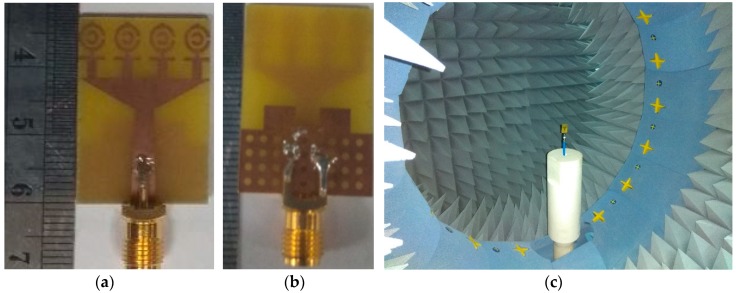
The photograph of the proposed ultrawideband (UWB) antenna (**a**) top view and (**b**) back view, and (**c**) UKM StarLab.

**Figure 9 sensors-18-04427-f009:**
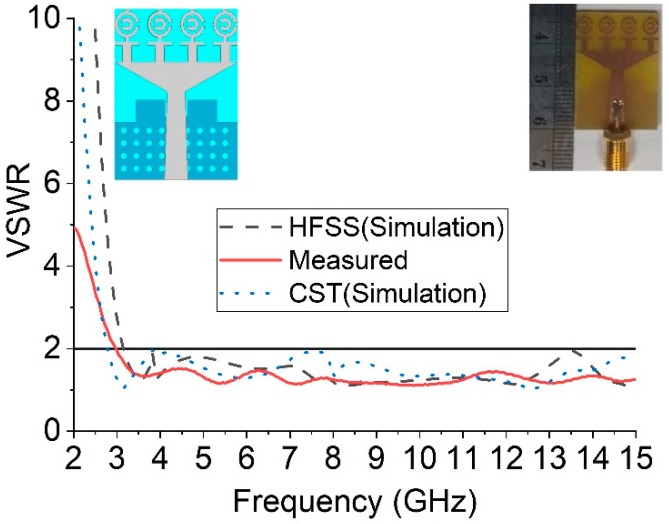
Measured and simulated VSWR curves.

**Figure 10 sensors-18-04427-f010:**
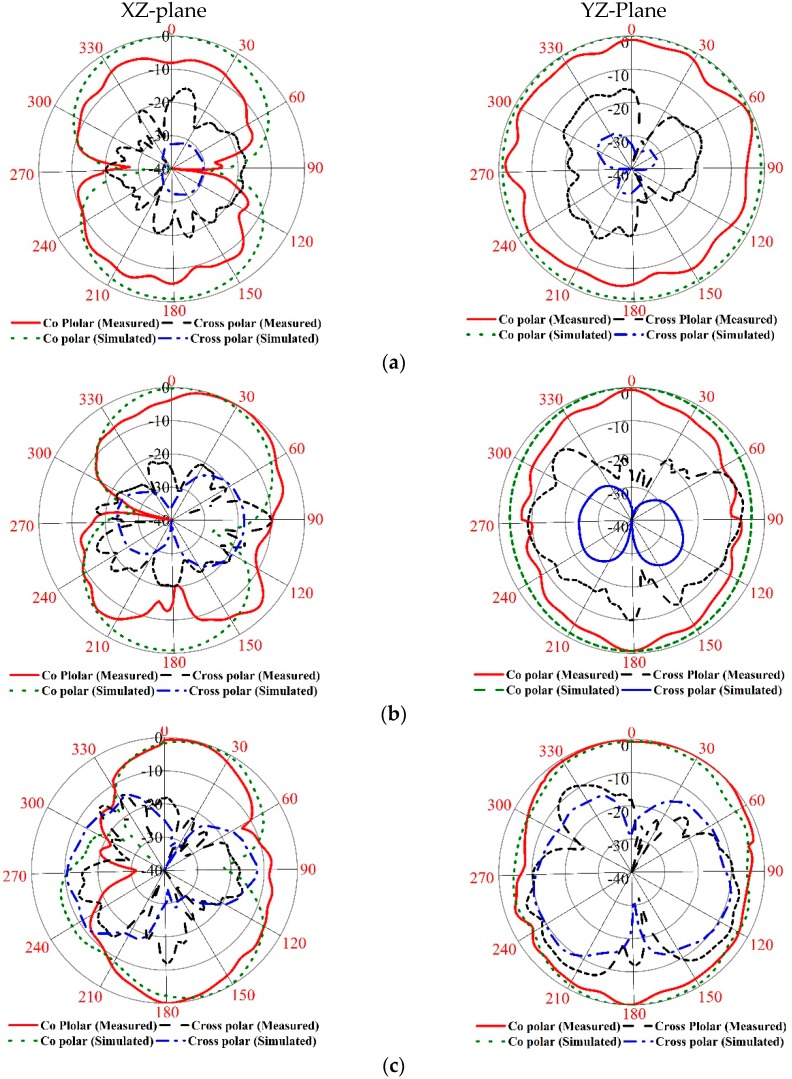
The 2D measured radiation pattern at (**a**) 3.25, (**b**) 6.5, and (**c**) 9.5 GHz.

**Figure 11 sensors-18-04427-f011:**
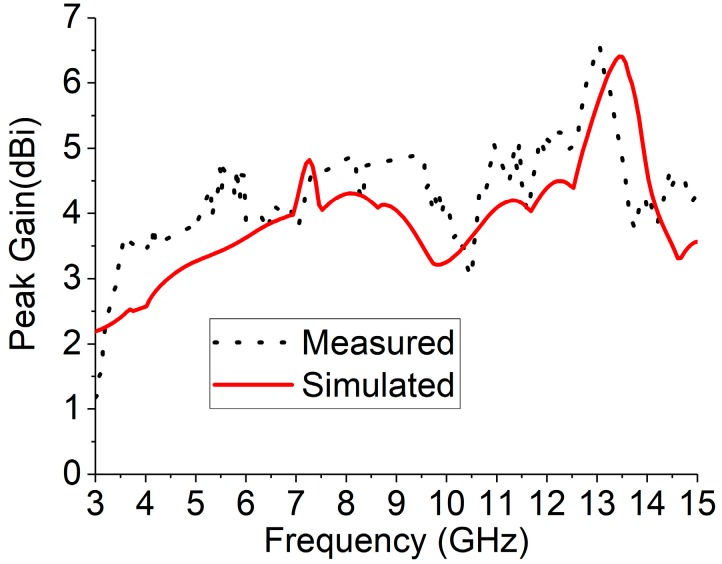
The simulated and measured gain of the antenna.

**Figure 12 sensors-18-04427-f012:**
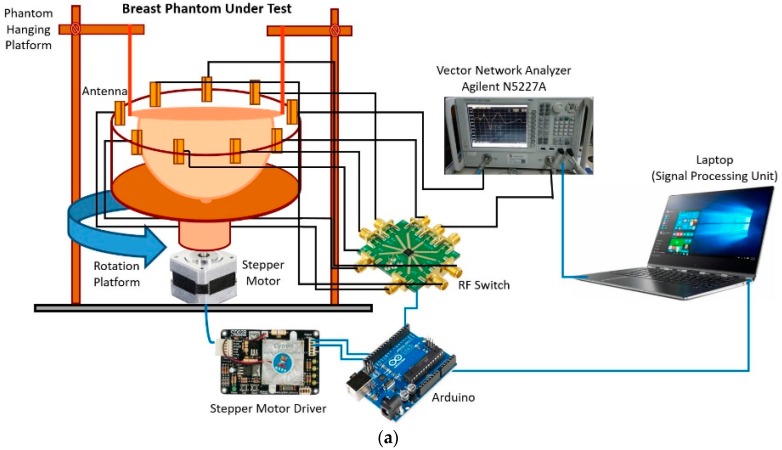
The architecture and a different component of the proposed experimental breast imaging system; (**a**) block diagram (**b**) breast phantoms.

**Figure 13 sensors-18-04427-f013:**
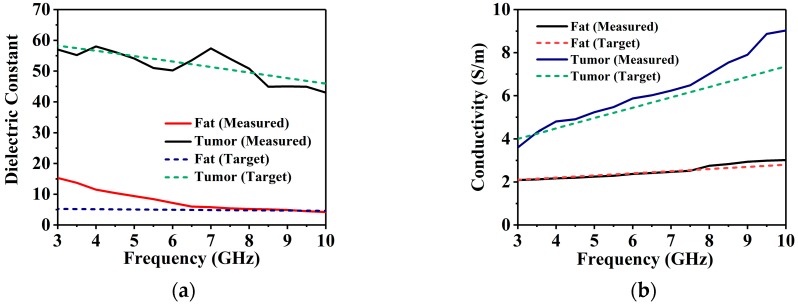
Dielectric constant and electrical conductivity of each material of homogenous (**a**,**b**), and heterogenous phantom (**c**,**d**).

**Figure 14 sensors-18-04427-f014:**
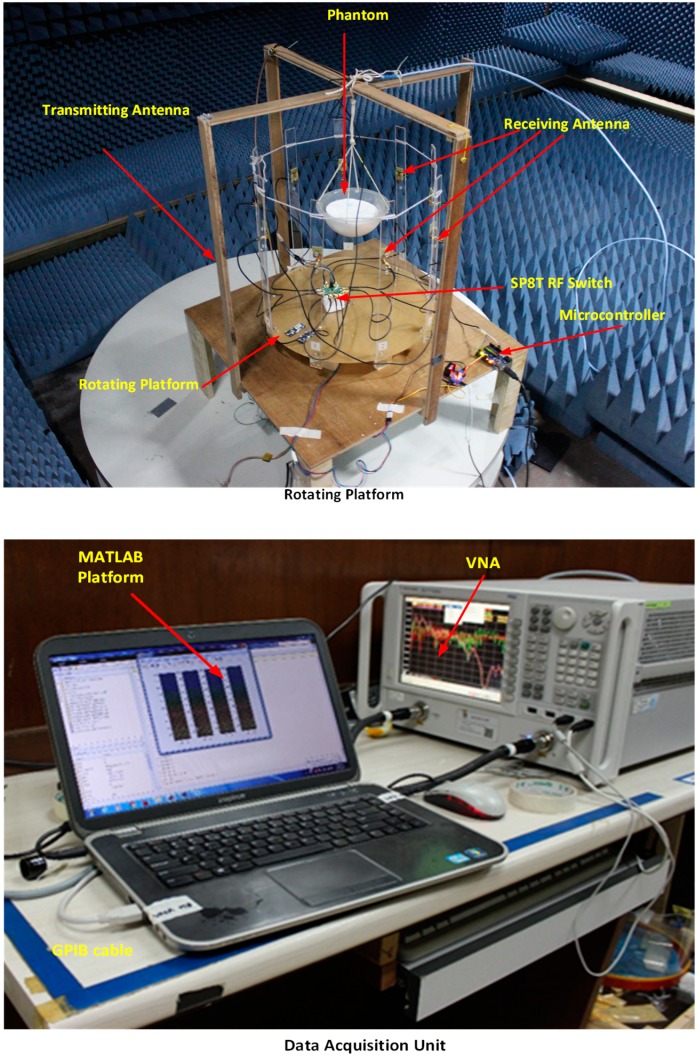
Experimental setup of imaging system.

**Figure 15 sensors-18-04427-f015:**
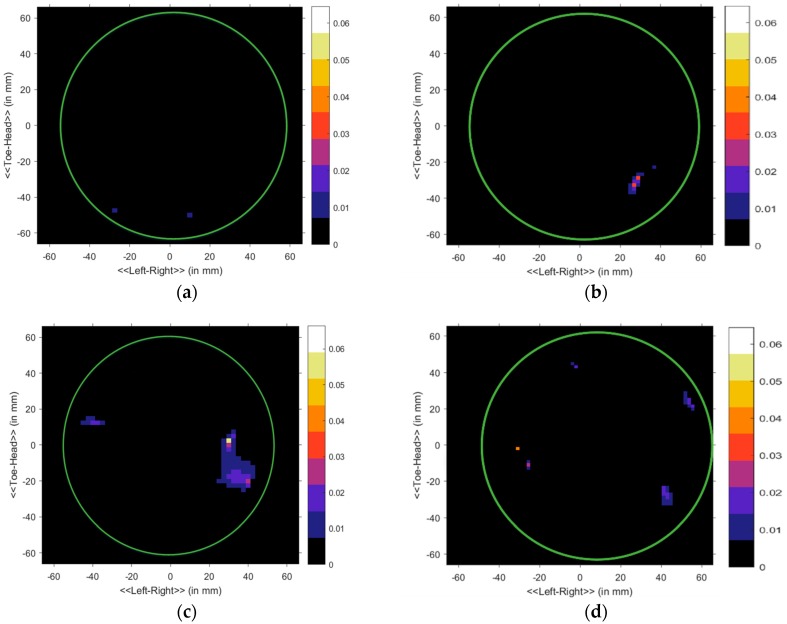
Imaging results. Homogenous: (**a**) no tumor, (**b**) 1 tumor, (**c**) 2 tumors, (**d**) 4 tumors; and heterogenous: (**e**) 2 tumors.

**Table 1 sensors-18-04427-t001:** The design parameters of the unit cell.

Parameter	Dimension (mm)	Parameter	Dimension (mm)
W_1_	3.95	g_2_	0.53
W_2_	0.48	g_3_	0.53
L_1_	8.25	c_1_	0.52
L_2_	2.64	c_2_	0.49
L_3_	0.53	r_1_	1
g_1_	0.53	r_2_	2

**Table 2 sensors-18-04427-t002:** The permeability, permittivity, and refractive index in the negative frequency zone.

Parameter	Negative Frequency Zone (GHz)
Permeability, *µ_r_*	6.7–12.10
Permittivity, *ɛ_r_*	2.8–4.36, 5.30–7.90, 9.50–10.25, 13.46–15.00
Refractive index, *n_r_*	7.36–10.48, 12.85–13.15

**Table 3 sensors-18-04427-t003:** Antenna design parameters according to Figure 4.

Parameter	Dimension (mm)	Parameter	Dimension (mm)
*W*	19.4	*W* _5_	4.7
*L*	27.5	*fw*	3.6
*L4*	9	*fw1*	2.8
*L5*	9.5	*fL*	13.9
*L6*	3.5	*g4*	3.6
*W* _3_	3.3	*g5*	2.13
*W* _4_	3.7	*h*	1.6
*R* _1_	0.5		

**Table 4 sensors-18-04427-t004:** Comparison of the existing imaging system with proposed system.

Investigator	Antenna Type	Operating Frequency (GHz)	Elements/Position	Fixed/Movable	Frequency/Time Domain	Imaging Method	Phantom and Tumor Object
[9]	Pyramidal Horn Antenna	2–10	8 × 241 scanned position on transmission reception	Movable multistatic	Frequency domain	No results	No No
[10]	Horn-like 3D UWB antenna	2–6.5	2 antenna elements 24 × 19 transmission reception position	Movable multistatic	Time and frequency domain	DMAS	No phantom No tumor
[13]	Balanced antipodal Vivaldi antenna	1–13	36 single element scanned position	Fixed tank rotate monostatic	Frequency domain	TSAR (tissue sensing adaptive radar)	Sample Tissue No
[14]	Corrugated antipodal Vivaldi antenna	1–4	16 single element scanned position	Fixed rotated platform monostatic	Frequency domain	DAS delay-and-sum algorithm)	Lab-based breast phantom No
[15]	Slotted antipodal Vivaldi antenna	3.01–11	Two element 2 × 50 position	Fixed platform rotated	Time domain	DMAS (delay-multiply-and-sum algorithm)	Yes Yes Single
[21]	Tapered and transmission loaded antenna	2–8	16 element array 16 × 15 scanned position	Fixed switching matrix	Time domain	DMAS	Lab-based breast phantom Yes Single
[16]	CPW feed monopole	2–4	16 elements array 16 × 15 scanned position	Fixed switching matrix	Time domain	DMAS	Yes Yes Single
[18]	Slotted patch	3.5–15	4 × 4 single element	Fixed switching matrix	Frequency domain	Confocal imaging	Simulated phantom Yes
[50]	UWB transceiver	3–10	16 element array	Fixed switching matrix	Time domain	DAS simulation only	No No No
[51]	CPW feed EBG structure antenna	3.1–7.6	2 antenna elements 2 × 120 scanned position	Fixed platform rotated	Frequency domain	DMAS	Commercial phantoms Single tumor object
Proposed	MTM unit cell loaded UWB patch antenna	3.0–8.0	9 MTM antenna array 8 × 50 scanned position	Movable multistatic	Frequency domain	DMAS	Lab-made phantom 1,2,4 tumor object

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
