# Peer review of "Experimental Breast Phantom Imaging with Metamaterial-Inspired Nine-Antenna Sensor Array"

_sensors, 2018, doi:10.3390/s18124427_

Reviewer 1 Report

General comments

The authors propose the design of a metamaterial based wide band antenna for breast cancer detection. Although the paper shows significant experimental facilities in the field of microwave breast imaging, from its reading is not easy to capture the focus and the novelty of the paper, and in particular whether it is related to the design of the proposed metamaterial antenna or to microwave imaging techniques, or neither one of them.

For this reason, the authors should emphasize the difference with their previous work [15], where another kind of antenna is proposed, but the same phantom and the same reconstruction algorithm has been substantially employed. Moreover, a deep comparison with works [37]-[38] is also required. 

Moreover, the DMSA approach, in my opinion cannot be reliably used for breast imaging since the anatomic structures of the breast are much more complicated with respect to those assumed in the paper (see paper by Lazebnik et al. and more recent papers exploiting enhanced dielectric or magnetic contrast agent microwave imaging). As a result, radar based approaches are not able to discriminate tumors, especially if they show similar properties to those of the fibroglandular and fibroconnective tissues and are very small in size. How the authors think to address this aspect of the problem?

My main suggestion is to resubmit the paper with the aim to enhance the scientific and/or technological novelty of the paper. 

Detailed comments

See the annotated pdf file

Author Response

Thank you very much for your valuable suggestions. Please find the enclosed revised manuscript. The revisions in the manuscript are marked in red color. In the revision, we have also addressed the  reviewers’ comments. The replies to the comments are attached here and marked  in green color.

Thanks

Reviewer 2 Report

The authors present an experimental system for breast phantom imaging.

They completely developed the single unit cell, the arising 4-elements antennas and the final array.

The novelty results in the adoption of metamaterials and in the reduction of antenna’s dimensions with respect to the state of the art. Finally, they tested the imaging system with lab-made phantoms and with already assessed imaging algorithm. 

The work is of potential interest for Sensors journal but some minor points need to be addressed before publication. 

First of all, I strongly encourage the authors to accurately review the writing of the paper as well as English form. Also the presentation of figures and equations should be improved (f.i., alignment of captions with figures, equation number in the same line with formulas, and so on).

Other minor concerns are listed below:

-       In line 59 the acronym MTS is used without a previous definition. Please define and use it thereinafter. 

-       Conversely, in line 79 the inversion algorithm “delay multiply and sum algorithm” is introduced without defining the acronym DMSA, that is instead used later. Please define the acronym in this point. Moreover, a reference paper for this method must be inserted. 

-       Please add the reference system in figure 1(b).

-       In line 114-115 the authors state “The proposed SRR structure magnetic resonance is better than the reported overlap and selp-oriented SRRs”. What does it means? The authors should motivate and/or give references about such a statement.

-       Which slab is referred to in line 122?

-       In line 141-142 the authors state “The used MTM unit cell and partial slotted ground plane are used for enhancing the impedance bandwidth with peak gain in the miniaturized antenna”. It should be motivated and/or properly referenced.

-       In Table 3, the dimension of the circular slot is not reported. 

-       Please improve the readability of figures 6 and 7 by adding the meaning of the horizontal dot-dashed black line in the label of figures. Please add also the measurement unit for y-axis in figure 7 and 12.

-       The authors present their antenna as a UWB antenna operating between 2.97-15GHz. However, they present figures 8 and 11 for low and middle frequencies. An additional figure for a higher frequency should be inserted.

-       Please add in figure 11 an explicit information about the considered planes.

-       Information in lines 331 and 332 are repeated. 

-       In line 338 a wrong information is given. In fact, in figure 14 the phantom surface is indicated by a continuous green line (not a dotted red line).

-       Are the inversion results of figure 14 qualitative or quantitative? Such a point is not sufficiently clear. Why the colorbar is not the same for the four images? This results in an inexplicable maximum scale white color. Finally, the phantom surface should be inserted for all images.

 Author Response

Thank you very much for your valuable suggestions. Please find the enclosed revised manuscript. The revisions in the manuscript are marked in red color. In the revision, we have also addressed the  reviewers’ comments. The replies to the comments are atatched here and marked  in green color.

Round  2

Reviewer 1 Report

As the study in the present paper is concerned with wideband systems (antennas, phantoms and radar based imaging) the authors should report both the dielectric constant and the losses (electrical conductivity) of each materials over the entire frequency band, for example measured curves and /or Debye or Cole-Cole fitting, or at least they should indicate the frequency which the above parameters are referred to.

Author Response

Dear Reviwer

Thank you very much for you suggestions. Please find the enclosed revised manuscript. The revisions in the manuscript are marked in red color. In the revision, we have addressed the  reviewers’suggestions. The replies to the comments are attached here.
